# Spontaneous activation of visual pigments in relation to openness/closedness of chromophore-binding pocket

Wendy Wing Sze Yue[1,2,3†‡], Rikard Frederiksen[4†], Xiaozhi Ren[1,2], Dong-Gen Luo[5,6,7,8,9], Takahiro Yamashita[10], Yoshinori Shichida[10], M Carter Cornwall[4], King-Wai Yau[1,2,11*]

[1]Solomon H. Snyder Department of Neuroscience, Johns Hopkins University School of Medicine, Baltimore, United States; [2]Center for Sensory Biology, Johns Hopkins University School of Medicine, Baltimore, United States; [3]Biochemistry, Cellular and Molecular Biology Graduate Program, Johns Hopkins University School of Medicine, Baltimore, United States; [4]Department of Physiology and Biophysics, Boston University School of Medicine, Boston, United States; [5]State Key Laboratory of Membrane Biology, Peking University, Beijing, China; [6]McGovern Institute for Brain Research, Peking University, Beijing, China; [7]Center for Quantitative Biology, Peking University, Beijing, China; [8]Peking-Tsinghua Center for Life Sciences, Beijing, China; [9]College of Life Sciences, Peking University, Beijing, China; [10]Department of Biophysics, Graduate School of Science, Kyoto University, Kyoto, Japan; [11]Department of Ophthalmology, Johns Hopkins University School of Medicine, Baltimore, United States

*For correspondence: kwyau@jhmi.edu

†These authors contributed equally to this work

Present address: ‡Department of Physiology, University of California, San Francisco, United States

Competing interests: The authors declare that no competing interests exist.

**Abstract** Visual pigments can be spontaneously activated by internal thermal energy, generating noise that interferes with real-light detection. Recently, we developed a physicochemical theory that successfully predicts the rate of spontaneous activity of representative rod and cone pigments from their peak-absorption wavelength ($\lambda_{max}$), with pigments having longer $\lambda_{max}$ being noisier. Interestingly, cone pigments may generally be ~25 fold noisier than rod pigments of the same $\lambda_{max}$, possibly ascribed to an 'open' chromophore-binding pocket in cone pigments defined by the capability of chromophore-exchange in darkness. Here, we show in mice that the $\lambda_{max}$-dependence of pigment noise could be extended even to a mutant pigment, E122Q-rhodopsin. Moreover, although E122Q-rhodopsin shows some cone-pigment-like characteristics, its noise remained quantitatively predictable by the 'non-open' nature of its chromophore-binding pocket as in wild-type rhodopsin. The openness/closedness of the chromophore-binding pocket is potentially a useful indicator of whether a pigment is intended for detecting dim or bright light.

## Introduction

Retinal rod and cone photoreceptors, although having similar phototransduction mechanisms, elaborate different morphological and molecular features for functioning in dim and bright light, respectively. At the pigment level, rod pigments have a low rate of spontaneous activation in darkness (*Baylor et al., 1980*), thus offering a good signal-to-noise ratio for dim-light vision. Spontaneous activation originates from internal thermal energy of the pigment molecule, generating an electrical event indistinguishable from that triggered by an absorbed photon (*Baylor et al., 1980*), thus interfering with real-light detection. Recently, *Luo et al. (2011)* have developed a macroscopic

**eLife digest** At the back of our eyes is a thin layer of cells that contain light-absorbing pigment molecules. These cells convert light energy into electrical signals that the brain then interprets to allow us to see. In this cell layer, the so-called cone cells work in bright light and provide us with the sense of color, whereas rod cells are for vision in dim light. Each visual pigment consists of a protein with a pocket-like space that holds a compound called a chromophore. Light causes the chromophore to change shape inside the pocket, which in turn activates the pigment. However, the pigments can also become activated at random, even in darkness. These false signals, nicknamed "dark light", are caused by heat instead of light and essentially create a kind of visual noise that can interfere with vision.

In 2011, researchers found that pigments that are most sensitive to the longer wavelengths of light (that is, light redder in color) tend to be noisier. The researchers also found that cone pigments are noisier than rod pigments even if they are most sensitive to the same wavelengths of light.

To understand what causes this difference between cone and rod pigments, Yue, Frederiksen et al. – who include many of the researchers involved in the 2011 study – made use of mice with a mutated pigment in their rod cells. The mutant pigment was more sensitive to light of shorter wavelengths and, importantly, it behaved like a cone pigment in some ways but kept the closed pocket that is found in rod pigments. Indeed, Yue, Frederiksen et al. showed that the noise level of this mutant pigment could be accurately predicted from the wavelength it was most sensitive to and how closed its pocket was (in other words, the pocket's "closedness"). Further analyses revealed that an open pocket seems to be common to cone pigments from different species. So, it appears that cone pigments are noisier because they have a more open pocket, and the extra space might allow the chromophore to move around and change shape more easily.

Going forward, more visual pigments need to be tested to confirm the relationship between the openness of the chromophore-binding pocket and spontaneous activity. If confirmed, it might be possible to one day predict whether a pigment is intended for dim- or bright-light vision simply by knowing whether its chromophore-binding pocket is more open or closed.

physicochemical theory about pigment noise based on the notion that a pigment's spontaneous activity originates from thermal isomerization, with an energy barrier closely related to the pigment's $\lambda_{max}$. By using multi-vibrational-mode statistical mechanics (*Ala-Laurila et al., 2004*; *Hinshelwood, 1940*; *St George, 1952*), the theory was able to explain quantitatively the $\lambda_{max}$-dependence of pigment noise, with the noise increasing by $10^7$-fold from blue (short-wavelength-sensitive, or SWS) cone pigment to red (long-wavelength-sensitive, or LWS) cone pigment (*Fu et al., 2008*; *Kefalov et al., 2003*; *Luo et al., 2011*). This theory clarifies the decades-long uncertainty about whether the spontaneous pigment activity arises from canonical isomerization of the pigment's chromophore (as in photoisomerization) or from some different, unknown chemical reaction.

Very interestingly, noise measurements in conjunction with the theory indicate that, for a given $\lambda_{max}$, a cone pigment may be generally ~25 fold more spontaneously active than a rod pigment (*Luo et al., 2011*). The simplest interpretation is that a cone pigment has a higher molecular frequency of attempting to cross the isomerization barrier (*Luo et al., 2011*). Concurrently, unlike rod pigment, a number of cone pigments show observable dark chromophore-exchange without isomerization when exposed to another chromophore (*Kefalov et al., 2005*; *Matsumoto et al., 1975*), suggesting a tendency of spontaneous dissociation between apo-cone-opsin and 11-*cis*-retinal by Schiff-base hydrolysis, in turn implicating the binding pocket being accessible – or 'open' – to external water. It was hypothesized that this 'openness' of cone pigments' chromophore-binding pocket – defined by the property of dark chromophore-exchange – imposes less constraint on the chromophore's attempts to isomerize spontaneously, resulting in a higher thermal noise compared to rod pigments for a given $\lambda_{max}$ (*Luo et al., 2011*).

Considering the fundamental success of the above theory in explaining the spontaneous activities of several representative rod and cone pigments, it is important to test the theory's overall predictive power more generally. However, this test is non-trivial, requiring in each case a separate genetic

mouse line expressing a test pigment for stringent interrogation in vivo. As such, the already-available $Rho^{E122Q/E122Q}$ knock-in mouse (*Imai et al., 2007*) offers an unusual opportunity. Its rods express a mutant rhodopsin with its Glu122 residue (conserved in rhodopsin) in the chromophore-binding pocket replaced by Gln, which is common in cone pigments. This E122Q mutation causes a blue-shift in $\lambda_{max}$ to ~480 nm from ~500 nm in wild-type (WT) rhodopsin (*Imai et al., 2007*), substantial enough for validating the quantitative connection between pigment noise and $\lambda_{max}$. Equally interestingly, this mutant rhodopsin has acquired some cone-pigment-like properties such as faster decays of the meta-II and meta-III states as well as a shift of the meta-I/meta-II equilibrium (*Imai et al., 2007*), although retaining the indication of a closed chromophore-binding pocket as gleaned from in vitro experiments (*Sakurai et al., 2007*). Thus, we can also check in this 'hybrid' pigment the correlation between pigment noise and openness/closedness of the chromophore-binding pocket as we hypothesized.

## Results and discussion

To examine spontaneous activation, we used $Rho^{E122Q/E122Q}$ mice in a $Guca1a^{-/-};Guca1b^{-/-}$ (more commonly known as $Gcaps^{-/-}$, and will be referred to as such) background, which removes the $Ca^{2+}$-dependent negative feedback on the guanylate cyclase via GCAP proteins in phototransduction (*Mendez et al., 2001*) and boosts the spontaneous event's amplitude by ~5 fold for easy identification over background noise. $Rho^{E122Q/E122Q};Gcaps^{-/-}$ rods had broadly similar morphology as $Rho^{WT/WT};Gcaps^{-/-}$ rods (*Figure 1A*). Expression levels of the pigment and other phototransduction components were also normal in mutant retinae based on Western blot analysis (*Figure 1B*). A normal expression level of the mutant pigment was further supported by electrophysiological measurements from single rods and by optical-density measurements by microspectrophotometry (Materials and methods). To measure pigment noise, we obtained ~10 min recordings in darkness from $Rho^{WT/WT};Gcaps^{-/-}$ and $Rho^{E122Q/E122Q};Gcaps^{-/-}$ rods (*Figure 1C*) and extracted the spontaneous-activation rate by two methods. The first was to count quantal events based on a criterion amplitude of >30% of the single-photon-response amplitude measured in the same cell and also on a criterion integration time (reflecting its overall kinetics) of being within 50–200% of that of the average dim-flash response (*Fu et al., 2008*). Collective data at 37.5°C gave 0.015 ± 0.010 $s^{-1}$ $cell^{-1}$ (mean ± SD, n = 12) for $Rho^{WT/WT};Gcaps^{-/-}$ rods, and 0.0024 ± 0.0025 $s^{-1}$ $cell^{-1}$ (n = 20) for $Rho^{E122Q/E122Q};Gcaps^{-/-}$ rods. The measurements from $Rho^{WT/WT};Gcaps^{-/-}$ rods matched previous estimates (*Burns et al., 2002*; *Fu et al., 2008*). A variation of this method (*Luo et al., 2011*) allows us to also validate the Poisson occurrence of spontaneous events, by dividing the dark records into 100 s epochs and counting the number of epochs containing no event, one event, two events, etc. Indeed, the resulting probability histogram fits the Poisson distribution (red lines in *Figure 1D*), with the probability, $p(u)$, of observing $u$ events in each epoch being given by $p(u) = w^u e^{-w}/u!$, where $w$ is the average number of events per epoch. From altogether 118 epochs from 20 rods, we obtained the $w$ value, giving a thermal rate of 0.0023 $s^{-1}$ $cell^{-1}$ for $Rho^{E122Q/E122Q};Gcaps^{-/-}$ rods at 37.5°C, very similar to the above measurement.

In the second method (*Fu et al., 2008*; *Kefalov et al., 2003*), we computed the power spectrum of the spontaneous events for each cell by subtracting the power spectrum of a segment in the dark recording with no obvious events from the power spectrum of the entire ~10 min recording. This 'difference spectrum' was then fitted with a scaled power spectrum of the same cell's average single-photon response (*Figure 1E*). From the scaling factor, we obtained a spontaneous-activation rate of 0.0025 ± 0.005 $s^{-1}$ $cell^{-1}$ (n = 11) for $Rho^{E122Q/E122Q};Gcaps^{-/-}$ rods at 37.5°C, very similar to that from the first method.

For a total of $6.5 \times 10^7$ rhodopsin molecules in a mouse rod (*Luo et al., 2011*), the overall molecular rate constant of spontaneous activation from either method above was $3.69 \times 10^{-11}$ $s^{-1}$, ~6-fold lower than WT ($2.31 \times 10^{-10}$ $s^{-1}$). From our theory (*Luo et al., 2011*), the predicted spontaneous-activation rate constant as a function of $\lambda_{max}$ is given by $Ae^{\frac{-0.84hc}{RT\lambda_{max}}} \sum_1^m \frac{1}{(m-1)!} \left(\frac{0.84hc}{RT\lambda_{max}}\right)^{m-1}$, where $h$ is Planck's constant, $c$ is velocity of light, $R$ is universal gas constant, $T$ is absolute temperature, and $m$ is the *nominal* number of vibrational modes in the pigment molecule contributing thermal energy to pigment isomerization. The pre-exponential factor $A$, taken to represent the frequency at which a pigment molecule attempts to isomerize thermally, was found in previous work (*Luo et al., 2011*) to

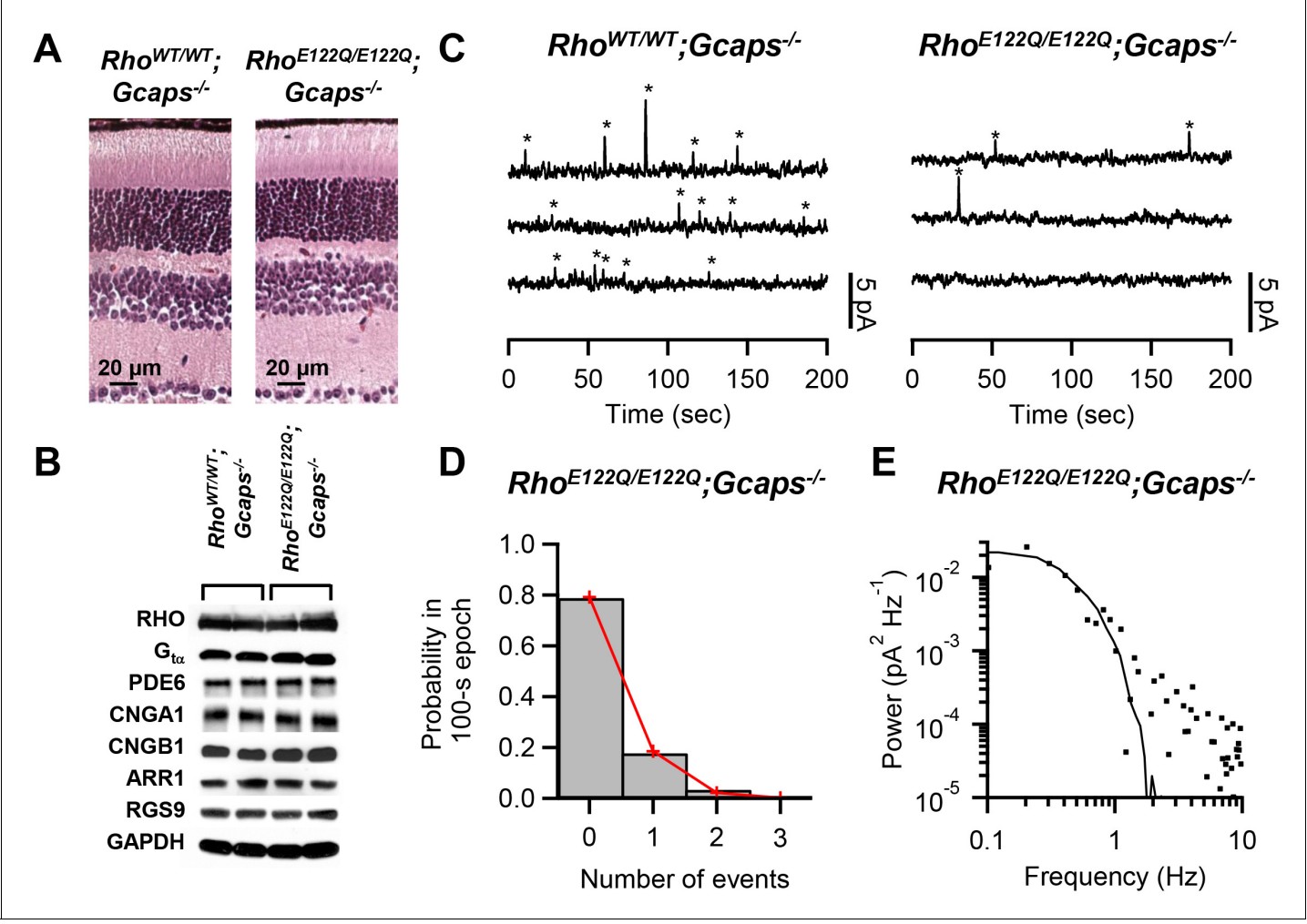

**Figure 1.** Measurement of spontaneous-activation rate of E122Q-rhodopsin. (**A**) Paraffin sections of 2.5-month-old $Rho^{WT/WT};Gcaps^{-/-}$ (left) and $Rho^{E122Q/E122Q};Gcaps^{-/-}$ (right) retinas stained by haematoxylin and eosin showing normal rod morphology. Similar results were found in altogether 3 sets of experiments. (**B**) Western blots from retinal extracts of $Rho^{WT/WT};Gcaps^{-/-}$(different animal in each of the left two columns) and $Rho^{E122Q/E122Q}$; $Gcaps^{-/-}$mice (different animal in each of right two columns) showing normal expression of various phototransduction protein components. RHO: rhodopsin; $G_{t\alpha}$: α subunit of transducin; PDE6: phosphodiesterase isoform 6; CNGA1: A1 subunit of cyclic nucleotide-gated (CNG) channel; CNGB1: B1 subunit of CNG channel; ARR1: Arrestin 1; RGS9: regulator of G protein signaling isoform 9; GAPDH: glyceraldehyde 3-phosphate dehydrogenase (control for protein amount). (**C**) Sample 10 min recordings from a $Rho^{WT/WT};Gcaps^{-/-}$ rod (left) and a $Rho^{E122Q/E122Q};Gcaps^{-/-}$ rod (right) in darkness. Traces (continuous from top to bottom) were low-pass filtered at 3 Hz. Quantal events were identified based on amplitude and kinetics (see Text) and are marked by asterisks. (**D**) Poisson analysis of dark recordings collected from all $Rho^{E122Q/E122Q};Gcaps^{-/-}$ rods. Bars indicate the measured probabilities of observing 0, 1, 2 and 3 events in 100 s epochs. A total of 118 epochs were analyzed. Red lines give the fit by the Poisson distribution with a mean event rate of 0.0023 $s^{-1}$ $cell^{-1}$. (**E**) Difference power spectrum (square symbols) of a $Rho^{E122Q/E122Q};Gcaps^{-/-}$ rod fitted with the power spectrum (curve) of the single-photon-response function.

The following source data is available for figure 1:

**Source data 1.** Source data for *Figure 1D*.

be $7.19 \times 10^{-6}$ $s^{-1}$ for rod pigments (with a 'closed' binding pocket) and $1.88 \times 10^{-4}$ $s^{-1}$ for cone pigments (with an 'open' binding pocket), a 26-fold difference. Inserting $T = (37.5 + 273)$ °K $= 310.5$ °K, $m = 45$ [see (*Luo et al., 2011*)], and $\lambda_{max} = 481$ nm for E122Q-rhodopsin (*Figure 2A*), we predict a molecular thermal-rate constant of $3.68 \times 10^{-11}$ $s^{-1}$ for a 'closed', and $9.63 \times 10^{-10}$ $s^{-1}$ for an 'open', binding pocket. Thus, the predicted rate for a 'closed' pocket matched the measurement very well. Recently, one of the authors here (Y.S.) has found biochemically a higher instead of lower

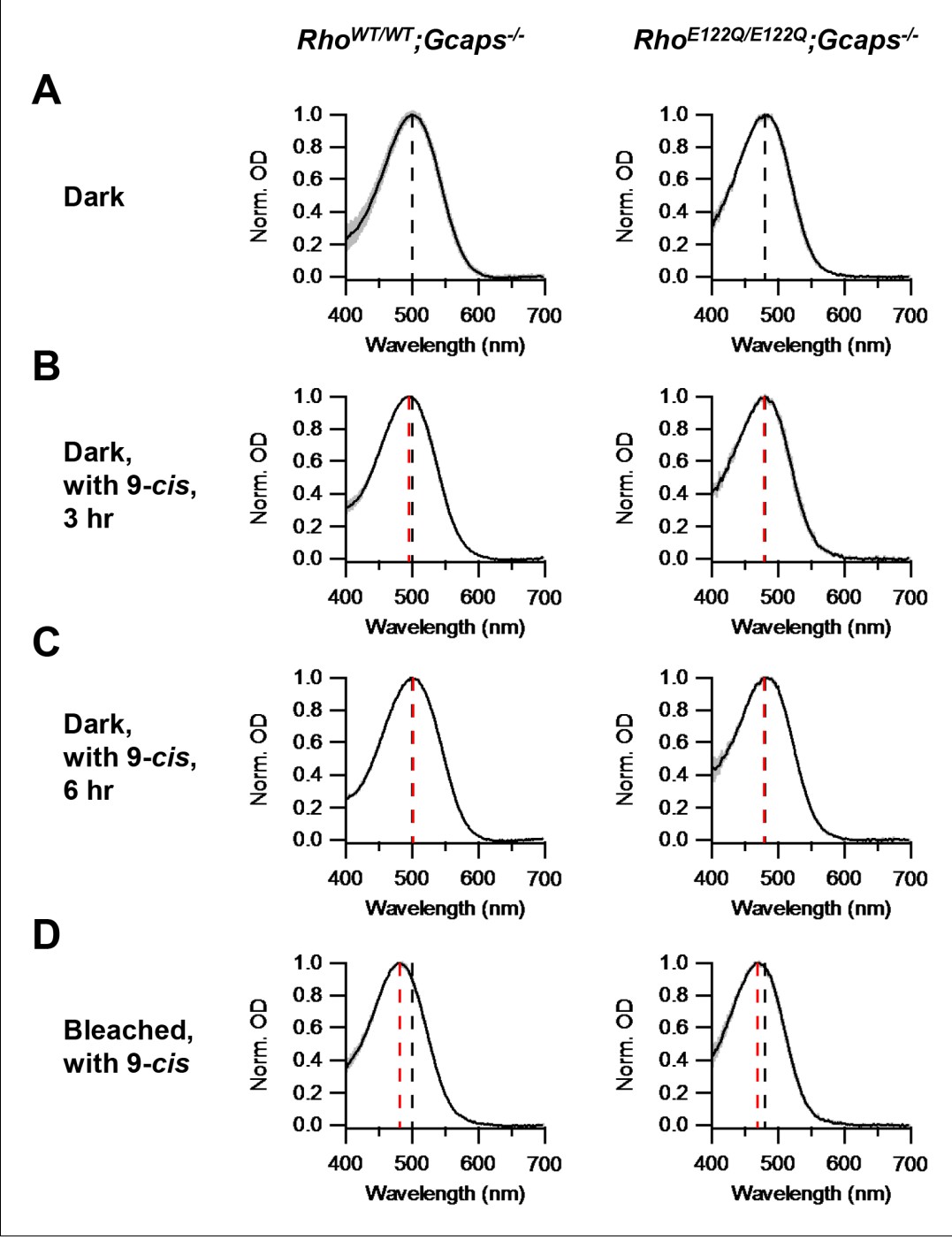

**Figure 2.** Chromophore-exchange experiment for probing the openness/closedness of chromophore-binding pocket. Absorption spectra (normalized to peak optical density) were obtained from dark-adapted $Rho^{WT/WT}$; $Gcaps^{-/-}$ rods (left) and $Rho^{E122Q/E122Q}$;$Gcaps^{-/-}$ rods (right) that were incubated in darkness in (**A**) Ames solution for 3 hr, (**B**) Ames solution with 15 µM 9-*cis*-retinal in darkness for 3 hr and (**C**) Ames solution with 15 µM 9-*cis*-retinal in darkness for 6 hr. (**D**) Absorption spectra from rods 99%-bleached followed by 3 hr 9-*cis* incubation in darkness. In all panels, curves are mean (black) ± SD (gray). Black dashed lines indicate the $\lambda_{max}$ of dark-adapted $Rho^{WT/WT}$;$Gcaps^{-/-}$ rods (left) and $Rho^{E122Q/E122Q}$;$Gcaps^{-/-}$ rods (right) not exposed to exogenous chromophore. Red dashed lines indicate the $\lambda_{max}$ of rods of the respective genotypes after the respective experimental treatment. The black dashed lines in (**A**) are replotted in (**B**), (**C**) and (**D**) for comparison with the red lines. For $Rho^{WT/WT}$; $Gcaps^{-/-}$ rods, $\lambda_{max}$'s are (**A**) 499.9 ± 4.8 nm (n = 33 recordings), (**B**) 495.6 ± 3.0 nm (n = 5 recordings, p=0.06), (**C**) 501.3 ± 4.5 nm (n = 6 recordings, p=0.52) and (**D**) 481.6 ± 4.1 nm (n = 5 recordings, p<0.0001), with p values from

*Figure 2 continued on next page*

*Figure 2 continued*

Student's t-test comparing with (**A**). For $Rho^{E122Q/E122Q}$;$Gcaps^{-/-}$ rods, $\lambda_{max}$'s are (**A**) 480.9 ± 5.4 nm (n = 7 recordings), (**B**) 479.6 ± 3.6 nm (n = 5 recordings, p=0.65), (**C**) 479.2 ± 3.9 nm (n = 5 recordings, p=0.56) and (**D**) 469.3 ± 3.2 nm (n = 8 recordings, p=0.0002), with p values from Student's t-test comparing with (**A**).

The following source data is available for figure 2:

**Source data 1.** Source data for *Figure 2.*

thermal rate constant of E122Q-rhodopsin over WT rhodopsin (*Yanagawa et al., 2015*). This discrepancy may arise from using detergent-solubilized samples in the biochemical method.

To investigate whether E122Q-rhodopsin indeed has a closed binding pocket, we checked its capability of chromophore-exchange in darkness. We incubated dark-adapted $Rho^{E122Q/E122Q}$; $Gcaps^{-/-}$ rods in Ames solution with or without 15 μM exogenous 9-*cis*-retinal (which, for a given opsin, forms a pigment with shorter $\lambda_{max}$ than does 11-*cis*-retinal) in darkness for 3 hr, then measured their absorption spectrum by microspectrophotometry (Materials and methods). No spectral shift was detected in the 9-*cis*-exposed rods, suggesting no dark chromophore-exchange, which is similar in behavior to $Rho^{WT/WT}$;$Gcaps^{-/-}$ rods (*Figure 2A,B*). We found no apparent exchange even with 6 hr of dark incubation (*Figure 2C*). As control, we delivered a 99%-bleach by 500 nm light to the rods prior to dark incubation with 9-*cis*. In this case, a spectral shift occurred in $Rho^{E122Q/E122Q}$; $Gcaps^{-/-}$ rods as in $Rho^{WT/WT}$;$Gcaps^{-/-}$ rods (*Figure 2D*), indicating normal hydrolysis and formation of the Schiff-base between E122Q-opsin and chromophore.

Chromophore-exchange experiments in live cells have previously been done only in salamander red cones (*Kefalov et al., 2005*). Given our hypothesis that the openness of a cone pigment's chromophore-binding pocket explains its higher spontaneous activity than that of rhodopsin of the same $\lambda_{max}$, we would like to check whether dark chromophore-exchange is indeed common also to other cone types. To avoid potential complications from in vitro conditions, we confined our question to live cells with microspectrophotometry, as described earlier and used previously on salamander red cones (*Kefalov et al., 2005*). We decided to examine zebrafish, which has reasonably large cones of multiple spectral types. *Figure 3A* shows their single-cell absorption spectra: red (LWS), green (medium-wavelength-sensitive, or MWS), blue (SWS), and ultraviolet-sensitive (UVS). After dark incubation with 15 μM 9-*cis*-retinal for 3 hr, the absorption spectra of dark-adapted, native (11-*cis*) LWS, MWS and SWS cones all shifted to shorter wavelengths, indicating incorporation of 9-*cis*-retinal (*Figure 3B*, with native spectra shown in black and spectra after 3 hr incubation with 15 μM 9-*cis*-retinal shown in red). Because pigments with shorter $\lambda_{max}$'s show smaller $\lambda_{max}$-differences between their 11-*cis*- and 9-*cis*-conjugated forms (the latter obtained by a full bleach followed by dark incubation with 9-*cis*; indicated in green in *Figure 3B*), the absolute amount of spectral shift due to dark chromophore-exchange was smaller for MWS and SWS cones than for LWS cones and was too small to be resolved for UVS cones. *Figure 3C* shows the time course of dark chromophore-exchange for LWS cones, quantified by the degree of spectral shift (Materials and methods). A single-exponential fit to the collected data (mean ± SD, 20–22°C) gives a time constant of 37 min, ~4 fold faster than previously found for salamander red cones (*Kefalov et al., 2005*), although the exchange in zebrafish did not appear to have reached completion (i.e.,<100%, see *Figure 3C*), possibly because of some 11-*cis*-retinal released from the pigment staying around and competing with the exogenous 9-*cis*-retinal. The time courses of dark chromophore-exchange for other zebrafish cone pigments were not well-resolved owing to the smaller spectral shift, but appeared to be faster at least for MWS and SWS cones, in that we found a significant shift to have occurred within the first 10 min of dark incubation (not shown). In contrast, zebrafish rhodopsin showed no obvious dark chromophore-exchange even after incubation with 9-*cis*-retinal for 6 hr (blue trace in *Figure 3B*, rod panel), same as mouse rhodopsin. Thus, at least for salamander and zebrafish, dark chromophore-exchange appears to be a general property for cone pigments with a time constant of the order of an hour, whereas rhodopsins behave quite differently. As such, we favor at present the simple approach of referring to these two collective groups of pigments as having 'open' versus 'closed' chromophore-binding pockets, respectively. If, in the future, the noise property and the pocket openness/closedness of a larger

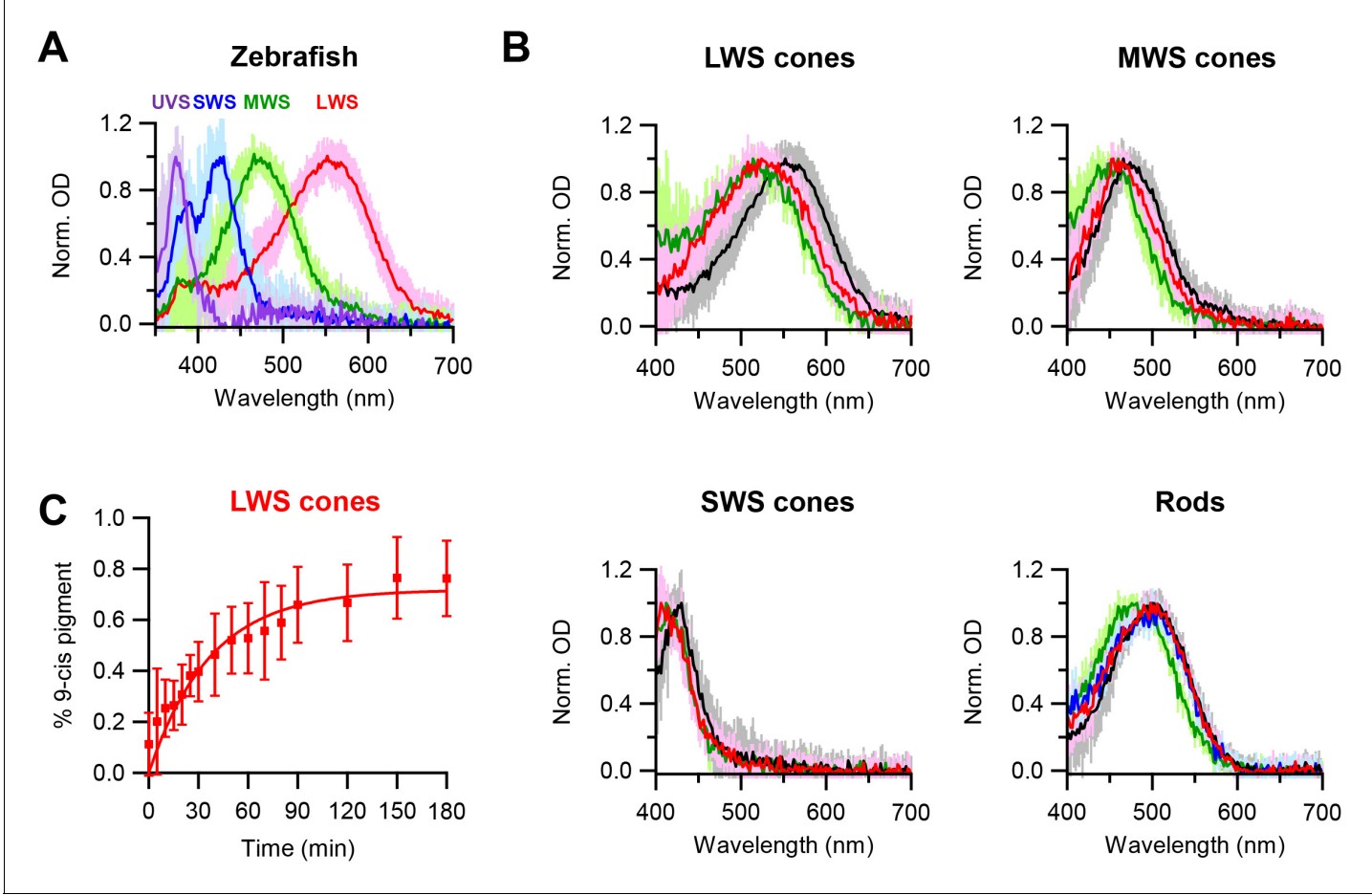

**Figure 3.** Chromophore-exchange experiment with zebrafish photoreceptors. (A) Absorption spectra of dark-adapted zebrafish long-wavelength-sensitive (LWS), medium-wavelength-sensitive (MWS), short-wavelength-sensitive (SWS) and ultraviolet-sensitive (UVS) cones. Spectra are mean (intense traces) ± SD (faint traces), normalized to peak optical density. n = 41 (LWS), 32 (MWS), 17 (SWS) and 5 (UVS) cells. (B) Absorption spectra of dark-adapted zebrafish rods and cones that were incubated with 15 μM 9-*cis*-retinal in darkness for 3 hr (red) and 6 hr (blue, only for rods). Spectra of dark-adapted cells after dark incubation for 3 hr in Ames solution without 9-*cis* (black), and of cells 99%-bleached followed by dark incubation for 3 hr in Ames solution with 9-*cis* (green), are given as reference. Spectra are mean (intense traces) ± SD (faint traces), normalized to peak optical density. For LWS cones, n = 24 (red), 41 (black) and 6 (green) cells. For MWS cones, n = 15 (red), 32 (black) and 7 (green) cells. For SWS cones, n = 10 (red), 17 (black) and 4 (green) cells. For rods, n = 13 (red), 16 (blue), 25 (black) and 20 (green) cells. (C) Time course of dark chromophore-exchange in LWS cones. The percentage of 9-*cis*-conjugated pigment (Materials and methods) is plotted (mean ± SD, n = 9 cells) against time in chromophore incubation. Curve is a saturating exponential function, $0.72 (1 - e^{-t/\tau})$, with an asymptote of 0.72 and a time constant, $\tau$, of 37 min fitted to the data.

The following source data is available for figure 3:

**Source data 1.** Source data for *Figure 3A*.
**Source data 2.** Source data for *Figure 3B*.
**Source data 3.** Source data for *Figure 3C*.

number of native and mutant pigments could be quantitatively measured, a finer sub-division may become more pertinent.

In summary, the E122Q-rhodopsin with hybrid rod- and cone-pigment-like properties has provided a quantitative validation and generalization of our macroscopic physicochemical theory of pigment noise previously proposed based on representative rod and cone pigments. At the same time, our hypothesized correlation between closedness/openness of the chromophore-binding pocket and pigment noise level also continues to hold. A low level of pigment noise is without exception

beneficial for dim-light vision. The successful application of our theory to E122Q-rhodopsin as a non-canonical pigment underscores the potential usefulness of evaluating a pigment as being intended for dim-light or bright-light function based on its noise level and, by extension, on the closedness/openness of its chromophore-binding pocket, provided the correlation between the two features continues to hold for other pigments. Such a functional criterion may be more informative than amino-acid-sequence comparison, especially for pigments with ambiguous evolutionary origin, such as the Gecko P467-pigment (*Kojima et al., 1992*).

### Note added in proof

A paper just appeared (*Tian et al., 2017*) reporting that rhodopsin purified from bovine rod outer segments dissociates into opsin and 11-cis-retinal in darkness, with a half life for holo-rhodopsin of the order of days. This rhodopsin behavior is not incompatible with our findings here because it is clearly still very different in time scale from the chromophore exchange in cone pigments, which occurs within hours.

## Materials and methods

### Animals

All animal experiments were carried out according to protocols approved by the Institutional Animal Care and Use Committee at Johns Hopkins University (MO14M199 for mouse) and Boston University (AN15427 for both mouse and zebrafish). Animals used in this study include $Rho^{WT/WT};Gcaps^{-/-}$ (RRID:MGI:3586516) and $Rho^{E122Q/E122Q};Gcaps^{-/-}$ mice as well as zebrafish (AB *Danio rerio*; RRID: ZIRC_ZL1).

### Histology

An eyeball of an acutely-euthanized animal was fixed in an alcohol-based zinc-formalin solution (Z-fix, Anatech, Battle Creek, MI) at room temperature overnight. The eyeball was then sent to the Johns Hopkins Medical Laboratories, where it was dehydrated through a series of increasing concentrations of ethanol, embedded in paraffin, and sectioned at a thickness of 5–8 μm. Sections close to the plane of the optic disc were collected, then de-paraffinized and rehydrated by passing through Xylene and a series of ethanol solutions of decreasing concentrations. After rinsing with water, the sections were stained with haematoxylin for 3 min. Following a wash with water, the sections were cleared, rinsed and blued. The sections were then rinsed again and stained with eosin for 1 min. Finally, the slides were rinsed, dehydrated through graded alcohols, cleared by Xylene and mounted.

### Western blotting

Retinas were isolated from euthanized mice into RIPA lysis buffer (140 mM NaCl, 0.1% Na-deoxycholate, 10 mM Tris-HCl, pH 8.0, 1 mM EDTA, 0.5 mM EGTA, 1% Triton X-100 and 0.1% SDS). Proteins were extracted by grinding the tissues with plastic pestles and vortexing every 5 min over a total of 30 min of incubation. Protein concentrations were determined using the bicinchoninic acid (BCA) Protein Assay Kit (Thermo Fisher Scientific, Waltham, MA). Subsequently, protein extracts (20 μg) were separated on 3–20% continuous SDS-PAGE gels (Bio-Rad, Hercules, CA) and transferred to polyvinylidene difluoride (PVDF) membrane. The membranes were blocked with 5% normal non-fat milk in TBST (500 mM NaCl, 20 mM Tris-HCl, pH 7.4, 0.1% Tween-20) for 1 hr and then incubated with different primary antibodies at 4°C overnight. Primary antibodies included a mouse anti-bovine rhodopsin (RHO) monoclonal antibody (1D4; 1:50; gift from Dr. Robert Molday, University of British Columbia), a rabbit anti-human transducin ($G_{t\alpha}$) polyclonal antibody (RRID:AB_2294749; 1:500; Santa Cruz, Dallas, TX), a mouse anti-bovine phosphodiesterase-6 (PDE6) monoclonal antibody (1: 1000; gift from Dr. Theodore Wensel, Baylor College of Medicine), a mouse anti-bovine cyclic-nucleotide channel subunit A1 (CNGA1) monoclonal antibody (PMc1D1; 1:100; gift from Dr. Robert Molday), a mouse anti-bovine CNG channel subunit B1 (CNGB1) monoclonal antibody (GARP4B1; 1:1000; gift from Dr. Robert Molday), a rabbit anti-mouse arrestin-1 (ARR1) polyclonal antibody (1:2500; gift from Dr. Jason C.-K. Chen), a rabbit anti-mouse regulator of G protein signaling isoform 9 (RGS9) polyclonal antibody (1:1000; gift from Dr. Jason C.-K. Chen), and a chicken anti-human

glyceraldehyde 3-phosphate dehydrogenase (GAPDH) polyclonal antibody (RRID:AB_10615768; 1:500; Millipore, Germany). After being washed with TBST, the blots were incubated with the appropriate HRP-conjugated secondary antibodies (1:10,000; Bio-Rad) at room temperature for 1 hr. Finally, the proteins on the membranes were detected by using the Enhanced Chemiluminescence (ECL) system (Thermo Fisher Scientific).

## Suction-pipette recording

One- to three-month-old mice were dark-adapted overnight, euthanized and their eyes removed under dim red light. The eyes were hemisected and the retinas were removed under infrared light in Locke's solution [112.5 mM NaCl, 3.6 mM KCl, 2.4 mM $MgCl_2$, 1.2 mM $CaCl_2$, 3 mM $Na_2$-succinate, 0.5 mM Na-glutamate, 0.02 mM EDTA, 10 mM glucose, 0.1% vitamins (Sigma-Aldrich, St. Louis, MO), 0.1% amino-acid supplement (Sigma-Aldrich), 10 mM HEPES, pH 7.4 and 20 mM $NaHCO_3$]. Retinas were stored in Locke's solution bubbled with 95% $O_2$/5% $CO_2$ at room temperature until use over not longer than 6 hr. When needed, a fraction of the retina was chopped into small pieces with a razor blade in the presence of DNase I (~20 U/ml) and was transferred to the recording chamber perfused with bubbled Locke's solution at 37.5°C ± 0.5°C. Temperature was monitored by a thermistor situated close to the recorded cell.

Single-cell recordings were made under infrared light by drawing the outer segment of a rod projecting from a fragment of retina into a tight-fitting glass pipette containing the following pipette solution: 140 mM NaCl, 3.6 mM KCl, 2.4 mM $MgCl_2$, 1.2 mM $CaCl_2$, 0.02 mM EDTA, 10 mM glucose and 3 mM HEPES, pH 7.4. In most experiments with light stimulation, 10- to 30-msec monochromatic flashes were used. Signals were sampled at 1 kHz through an Axopatch 200B amplifier and low-pass filtered at 20 Hz (RC filter, Krohn-Hite, Brockton, MA), unless specified otherwise.

## Measurements of the rates of spontaneous activation

The average single-photon response function [$f(t)$] was computed by first obtaining the average response profile of a rod to 80–100 identical dim flashes and then scaling it to the amplitude of the single-photon response, which was calculated as the ensemble variance-to-mean amplitude ratio at the transient peak of these dim-flash responses.

Continuous 10 min recordings were obtained from rods in complete darkness, and the rate constant of spontaneous activation were measured by two methods. In the direct counting method, traces were usually low-pass filtered at 3 Hz for identifying and counting quantal events. Two criteria were imposed during identification: (1) the amplitude of the event should be >30% of the single-photon response amplitude of the same cell, and (2) the integration time of the event should be within 50–200% of that of the average dim-flash response. The cellular rate constant of thermal activation was given by the total number of spontaneous events divided by the total recording time for each cell. Alternatively, the dark-recording traces were divided into 100 s epochs. The frequency of observing no event, one event, two events, etc. in an epoch was plotted and fitted with the Poisson distribution $p(u) = w^u e^{-w}/u!$, where $p(u)$ was the probability of observing $u$ events in each epoch and $w$ was the average number of spontaneous activation event per 100 s epoch.

In the second method, power spectra were computed from the entire dark-recording trace and from a segment of it containing no obvious spontaneous events (based on visual inspection) for each cell by using Clampfit 9 (Molecular Devices, Sunnyvale, CA) in 8.192 s segments with 50% overlap. The difference spectrum between these two spectra constituted the spectrum for the spontaneous events. This difference spectrum was fitted with a scaled power spectrum of the average single-photon response function [$f(t)$; see above] of the same cell. The rate of spontaneous isomerization is given by the scaling factor divided by the acquisition time (8.192 s).

The molecular rate constant was obtained by dividing the measured cellular rate by the number of pigment molecules ($6.5 \times 10^7$) per rod. The expression of rhodopsin appeared normal in $Rho^{E122Q/E122Q};Gcaps^{-/-}$ retinas based on Western blotting (**Figure 1B**). Another way to assess pigment content in a rod outer segment is to measure the probability ($p_s$) of successfully eliciting electrical responses in an experiment using repeated dim-flash trials of known intensity. This probability is related to the rod outer segment's effective collecting area ($A_e$) and the flash intensity ($I$) by $p_s = 1 - e^{-A_e I}$. In turn, $A_e$ is directly proportional to the pigment content. As such, we found $A_e$ to be 0.44 ± 0.09 $\mu m^2$ (mean ± SD, n = 10) for $Rho^{WT/WT};Gcaps^{-/-}$ rods and 0.35 ± 0.10 $\mu m^2$ (mean ±

SD, n = 18) for $Rho^{E122Q/E122Q}$;$Gcaps^{-/-}$ rods. Thus, the E122Q/WT pigment-content ratio is 1/1.26. Meanwhile, microspectrophotometry (see below) measured an average relative peak optical density of 0.37 ± 0.07 unit (mean ± SD, n = 44 recordings from seven experiments) for $Rho^{E122Q/E122Q}$; $Gcaps^{-/-}$ rods and 0.30 ± 0.09 unit (n = 84 recordings from 33 experiments) for $Rho^{WT/WT}$;$Gcaps^{-/-}$ rods, giving a E122Q/WT pigment-content ratio of 1.23/1. The mild discrepancy between methods may reflect measurement uncertainties. Taken together, the pigment levels appear similar between $Rho^{WT/WT}$;$Gcaps^{-/-}$ and $Rho^{E122Q/E122Q}$;$Gcaps^{-/-}$ rods.

## Pigment noise prediction

The rate of spontaneous activation (*k*) is given by:

$$k = A e^{\frac{-0.84hc}{RT\lambda_{max}}} \sum_1^m \frac{1}{(m-1)!} \left( \frac{0.84hc}{RT\lambda_{max}} \right)^{m-1},$$ (1)

where *A* is the pre-exponential factor, *h* is Planck's constant ($1.58 \times 10^{-37}$ kcal sec), *c* is speed of light ($3.00 \times 10^{17}$ nm sec$^{-1}$), *R* is universal gas constant ($1.99 \times 10^{-3}$ kcal °K$^{-1}$ mol$^{-1}$), *T* is absolute temperature (310.5 °K) and *m* is the nominal number of vibrational modes contributing thermal energy to pigment activation. Based on previous work (*Luo et al., 2011*), *m* is 45 for rhodopsin and is taken to be the same for cone pigments, given the same chromophore. The average *A*-values were empirically determined to be $7.19 \times 10^{-6}$ s$^{-1}$ for rod pigments and $1.88 \times 10^{-4}$ s$^{-1}$ for cone pigments (*Luo et al., 2011*). Predictions were made by substituting these parameters and $\lambda_{max}$ = 481 nm (for E122Q-rhodopsin) into *Equation 1*.

## Microspectrophotometry

For experiments on mouse rods, mice were dark-adapted for 12 hr before experiment. After euthanization, eyes were removed under dim red light. Under infrared illumination, the eyes were hemisected and the retinas were isolated in HEPES (10 mM, pH 7.4)-buffered Ames medium (Sigma-Aldrich). Each retina was divided in half, yielding altogether four pieces of tissues to be subjected to different treatments. Two pieces of retina were kept dark-adapted and incubated for 3 hr in darkness in HEPES-buffered Ames medium containing 1% fatty-acid-free bovine serum albumin (BSA) with or without 15 µM 9-*cis*-retinal; the other two pieces of retina were subjected to a 99%-bleach (see below) and then incubated in the same HEPES-buffered, BSA-supplemented Ames medium as above with or without 15 µM 9-*cis*-retinal.

After their respective treatments, the absorbance spectra of the retinal pieces were measured using a custom-built microspectrophotometer. A retinal piece was gently flattened by forceps and a slice anchor (Warner Instruments, Hamden, CT) on a quartz cover-slip window in the bottom of a 2 mm-deep Plexiglass recording chamber with the photoreceptors facing up. The recording chamber was placed on a microscope stage located in the beam path of the microspectrophotometer. The retinal tissue was superfused at a rate of 4 ml/min with Ames medium (Sigma-Aldrich) buffered with sodium bicarbonate and equilibrated with 95% $O_2$/5% $CO_2$. Temperature was maintained at 35–37°C. Absorption spectra were obtained from a region of the retina along its edge where outer segments could be seen protruding perpendicular to the light beam, with tens of outer segments in the light path. The measured area contained predominantly rod instead of cone photoreceptors, as evinced by the $\lambda_{max}$. Measurements were made over the wavelength range of 300–700 nm with 2 nm resolution, with the polarization of the incident beam parallel to the plane of the intracellular disks (T-polarization). The absorbance spectrum was calculated from Beers' Law $OD = log(I_i/I_t)$, where *OD* is the optical density or absorbance, $I_i$ is the light transmitted through a cell-free space adjacent to the outer segments, and $I_t$ is the light transmitted through the tissue. Generally, 10 complete sample scans and 10 baseline scans were averaged to increase the signal-to-noise ratio. All absorbance spectra were baseline-corrected.

For experiments on zebrafish photoreceptors, wild-type (AB) zebrafish (*Danio rerio*), obtained from the colony held by the Animal Science Department at Boston University School of Medicine, was dark-adapted for 12 hr prior to experimentation. Euthanasia, dissection and tissue manipulation were performed in darkness with the aid of infrared image converters. Fishes were euthanized by exposure to cold (0°C) water followed by decapitation. The eyes were removed and hemisected in recording solution containing 104 mM NaCl, 2.5 mM KCl, 1.2 mM MgCl$_2$, 1.6 mM CaCl$_2$, 0.1 mM

NaHCO$_3$, 1 mM-NaH$_2$PO$_4$, 1 mM sodium pyruvate, 15 mM glucose, 15 mM HEPES (acid), 5 mM HEPES (base, Na-salt), 0.5 µg/ml insulin, 5 µg/ml d-biotin, 70 µl/ml fetal bovine serum, 10 µl/ml penicillin streptomycin, 150 µg/ml L-glutamine, 10 µL/ml 50× MEM amino acids, 5 µl/ml 100× MEM vitamins, pH = 7.8. The retinas were then isolated from the eyecups and the retinal pigment epithelium. Retinal tissues not immediately used for experiment were stored in the above solution in a dark container on ice.

For experiments in *Figure 3B,* a retina was treated off-stage in one of the following four ways: (1) directly used for microspectrophotometric measurements to obtain dark spectra, (2) incubated in recording solution with additional 1% bovine serum albumin (fatty-acid-free) and 15 µM 9-*cis*-retinal in darkness for 3 hr, (3) incubated with 1% bovine serum albumin and 15 µM 9-*cis*-retinal in the same way for 6 hr, and (4) bleached (see below) and incubated with 1% bovine serum albumin and 15 µM 9-*cis*-retinal in the same way for 3 hr. After treatment, the retina was cut into small (~50 µm×50 µm) pieces and then triturated in solution, producing isolated photoreceptors. Cells were transferred to a recording chamber containing recording solution maintained at 20–22°C. Different types of photoreceptors were identified by their morphology and confirmed by spectral absorbance, which was recorded similarly as in mouse experiments except from single zebrafish photoreceptors.

To measure the time course of chromophore-exchange in zebrafish LWS cones (*Figure 3C*), dark-adapted photoreceptors were dissociated as above directly after retina isolation and were transferred to the MSP recording chamber. A LWS cone was identified and its dark spectrum measured. The solution in the recording chamber was then replaced by recording solution containing 1% bovine serum albumin and 15 µM 9-*cis*-retinal. Measurements of spectral absorbance were made periodically over 3 hr, at 20–22°C. The light for probing the spectrum at a given time point was at an intensity that would bleach less than 0.1% of the pigment content per scan. To quantify the degree of chromophore-exchange, we used the spectrum of dark-adapted LWS cones (*Figure 3B*, black) and that of bleached LWS cones regenerated with 9-*cis*-retinal (*Figure 3B*, green) as the spectra for 11-*cis*- and 9-*cis*-pigment, respectively. A polynomial with degree 10 was fitted into each of these spectra. For each LWS cone, the spectrum acquired at each time point during 9-*cis* incubation was fitted in the 510–750 nm range with a linear combination of the two polynomials to obtain the percentage of 9-*cis*-conjugated pigment. Data from all cells were then averaged.

## Pigment-bleaching

For mouse rods, bleaching was performed off-stage on a portable optical bench consisting of a tungsten/halogen lamp, a set of neutral density filters, a 500 nm interference filter and a small aperture (3.25 mm). The retinal tissue was placed in Ames medium in a 35 mm petri dish under the focused circular light spot. The onset of light was controlled by a manual shutter. The bleached fraction, $F$, was estimated from the relation $F = 1 - e^{-IPt}$, where $I$ was the bleaching light intensity ($1.33 \times 10^6$ photons µm$^{-2}$ s$^{-1}$), $P$ was the photosensitivity [$5.7 \times 10^{-9}$ µm$^2$; see (*Woodruff et al., 2004*)] of mouse rhodopsin measured in situ at its $\lambda_{max}$ and $t$ was the duration of light exposure; the retinal tissue was typically light-exposed for 16 min to achieve a > 99% bleach. For zebrafish photoreceptors, bleaching was done as above except for using recording solution instead of Ames solution, and using white light of the same source intensity (i.e., not including the 500 nm interference filter).

## Chromophore-preparation

9-*cis*-retinal was handled in dim red light. A stock solution of 30 mM 9-*cis*-retinal was prepared by dissolving 9-*cis*-retinal in ethanol. The peak absorbance (OD) of retinoid in the stock solution was measured using a conventional spectrophotometer, and its concentration was calculated as $c = (OD_{373}l)/\varepsilon_{373}$, where $l$ was a 1 cm path length and $\varepsilon_{373} = 36,100$ M$^{-1}$ cm$^{-1}$ was the extinction coefficient of 9-*cis*-retinal in ethanol. Working solutions containing 9-*cis*-retinal were prepared by first adding 1 µl of stock solution to a conical vial. HEPES-buffered Ames medium containing 1% delipidated BSA was then added in multiple times in increasing amounts ($9 \times 5$ µl, $1 \times 50$ µl, $2 \times 450$ µl, $1 \times 1000$ µl) until the final volume was 2 ml; the concentration of 9-*cis*-retinal in the working solution was 15 µM.

## Acknowledgements

We thank Dr. Jeannie Chen (University of Southern California) for the *Gcaps*$^{-/-}$ line. We also thank Drs. Robert Molday (University of British Columbia), Jason C-K Chen and Theodore Wensel (Baylor College of Medicine) for antibodies. This work was supported by NIH Grants EY06837 (K-WY) and EY001157 (MCC); National Natural Science Foundation of China Grant 31471053 (D-GL); the António Champalimaud Vision Award, Portugal (K-WY); and a Howard Hughes Medical Institute International Predoctoral Fellowship (WWSY).

## Additional information

### Funding

| Funder | Grant reference number | Author |
| --- | --- | --- |
| National Eye Institute | EY06837 | King-Wai Yau |
| National Eye Institute | EY001157 | M Carter Cornwall |
| National Natural Science Foundation of China | 31471053 | Dong-Gen Luo |
| Antonio Champalimaud Foundation, Portugal | Antonio Champalimaud Vision Award | King-Wai Yau |
| Howard Hughes Medical Institute | International Student Research Fellowships | Wendy Wing Sze Yue |

The funders had no role in study design, data collection and interpretation, or the decision to submit the work for publication.

### Author contributions

WWSY, Conceptualization, Resources, Data curation, Formal analysis, Validation, Investigation, Visualization, Methodology, Writing—original draft, Writing—review and editing; RF, Data curation, Formal analysis, Validation, Investigation, Visualization, Methodology, Writing—review and editing; XR, Data curation, Formal analysis, Investigation, Methodology; D-GL, Resources, Supervision, Methodology; TY, YS, Resources, Validation, Investigation; MCC, Resources, Data curation, Formal analysis, Supervision, Funding acquisition; K-WY, Conceptualization, Resources, Formal analysis, Supervision, Funding acquisition, Validation, Writing—original draft, Project administration, Writing—review and editing

### Author ORCIDs

King-Wai Yau, http://orcid.org/0000-0001-8880-3091

### Ethics

Animal experimentation: All animal experiments were carried out according to protocols approved by the Institutional Animal Care and Use Committee at Johns Hopkins University (MO14M199 for mouse) and Boston University (AN15427 for both mouse and zebrafish).

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
