## [Decision Letter]

Thank you for submitting your article "Spontaneous Activation of Visual Pigments In Relation To Openness of Chromophore-Binding Pocket" for consideration by *eLife*. Your article has been reviewed by three peer reviewers, and the evaluation has been overseen by Richard Aldrich as the Senior and Reviewing Editor. The following individuals involved in review of your submission have agreed to reveal their identity: Theodore G Wensel (Reviewer #3).

The reviewers have discussed the reviews with one another and the Reviewing Editor has drafted this decision to help you prepare a revised submission.

The reviewers are generally enthusiastic about the work, but they offer several criticisms and suggestions for improvement. Ordinarily we excerpt the most critical comments for the focused attention of the authors, but in this case the reviewers and editor felt you should see their full reviews to help in your revisions.

*Reviewer #1:*

Yue et al. here attempt to test an earlier "macroscopic physicochemical theory" (Luo et al., 2011) concerning the basis for thermal noise in visual photoreceptor cells based on an analysis of molecular pigment type (rod versus cone) and wavelength of maximal absorption (λ_max_). The primary experimental system is a knockin/knockout mouse (*Rho^E122Q/E122Q^, Gcaps^-/-^*) in which the usual wild-type rhodopsin (Rho) is replaced by the mutant Rho E122Q, which has been characterized in some detail earlier using a variety of methods. Much of the paper seems to be a reiteration of the earlier 2011 paper and the authors repeatedly refer to "our" theory or model even though it appears that only 2 of the 8 current authors were associated with the earlier work. Although this paper ascribes to being quantitative in presenting Rho E122Q is a definitive experimental example to validate the earlier theory paper, the claims of the paper are either too vague (for example, using ill-defined terms like "openness" of the retinal binding pocket), or too overreaching (for example, claiming that a single test case proves the earlier theory. In fact, it's not clear that the Rho E122Q mouse is a logical or sufficient model system for a variety of reasons. However, the data set is interesting and valuable and there are no major issues, but the authors should consider rewriting sections of the paper and should address a number of questions/concerns as outlined below:

1) Abstract. "…spontaneously activated by internal thermal energy" – what does that mean, actually? Isn't is just thermal activation? "Confusion in the field"? What is the confusion? "Quantitatively correlated with the closedness of its chromophore-binding pocket"? How is closedness actually quantitated? It isn't, so any correlation cannot be quantitative. Is it the "closedness" or the λ_max_ value in the end that's the most important factor. The Abstract is extremely confusing in and of itself.

2) Introduction. Of course the ground-state isomerization energy barrier is closely related to the λ_max_. That's trivial, isn't it? Why is that a part of a unifying theory? The trivial definition of "open" pocket refers to accessibility of the Schiff base to hydrolysis, for example by hydroxylamine (with a 1955 Wald reference), but the actual chemistry of the hydrolysis reaction is a very complicated pH dependent mechanism that is only partly dependent on accessibility or hydration state. It is not stated why the E122Q pigment is cone-like, or "non-canonical." Some detail is required to justify its use, other than the mouse model happened to exist. What other mutants might be useful and why weren't they used or considered?

3) Results, third paragraph. What is the total number of E122Q pigment molecules per cell? Can it be assumed that it is the same as the total Rho in WT? If so, why? The scaling factor related to the number of molecules is very important in the equation used to a spontaneous activation rate constant. Also at the end of the subsection “Measurements of the rates of spontaneous activation”, it is stated that the "molecular rate constant could be obtained by further dividing the measured cellular rate by the number of pigment molecules per rod." Again, how is it assumed that E122Q is the same as Rho?

4) Results, last paragraph. The competition experiment seems problematic. Why was only a single time point of 3 hours in darkness chosen? Does the retinal bind to the 1% BSA predominantly (subsection “Chromophore-preparation”, Methods)?

5) Results, end of last paragraph. It is now stated that E122Q is resistant to hydroxylamine? What? Isn't the λ_max_ of 9-cis Rho 490 nm, not 482 nm? The data actually show Rho at about 498 nm or so, not 500. Why not use the actual data values in the text?

6) Discussion. The main criticism is that the authors need to justify how one example case can provide a justification for the earlier theory, and why and how the E122Q Rho is a legitimate test system. Does it really have hybrid rod-cone properties when expressed in a mouse rod cell?

7) Subsection “Pigment noise prediction”. m is taken to be the same for Rho and cone pigments? That's really strange and some additional explanation seems warranted.

8) Figure 1. Again, referring to the issue of E122Q quantitation, the Rho band looks much heavier than the E122Q knockin band.

*Reviewer #2:*

This manuscript entitled "Spontaneous Activation of Visual Pigments In Relation To Openness of Chromophore-Binding Pocket" by Yue, Frederiksen, Ren, Luo, Yamashita, Shichida, Cornwall & Yau tests the validity of a previously proposed model (Luo et al., 2011) correlating the spectral sensitivity of visual pigments with their thermal stability. This model is currently the dominant unifying theory correlating the two key functional properties of visual pigment molecules: their spectral sensitivity characterized by the peak-absorption wavelength (λ_max_) and their susceptibility to thermal noise characterized by the rate of thermal isomerizations. The theory by Luo et al. proposes a similar dependency between the dark event rate and λ_max_ for rod and cone pigments. However, cone pigments with the same λ_max_ as rod pigments have overall 25-fold higher dark even rates according to measurements. In the model formulation, this has been implemented by using a pre-exponential factor that is 25-fold larger for cone pigments compared to rod pigments. This difference has been earlier hypothesized to relate to the closeness/openness of the chromophore-binding pocket (Ala-Laurila et al., 2004 & Luo et al., 2011). The validity of this hypothesis has not been tested directly experimentally. The current paper tests the validity of this interpretation. The current paper uses a mutant rhodopsin (E122Q-rhodopsin) with some cone-like pigment properties (e.g. faster decay of rhodopsin intermediates, meta-II and meta-III) but with a rod-like chromophore exchange rate (correlated with the openness of the binding pocket). The paper shows that the dark-event rate of E122Q-rhodopsin is in accordance with the model prediction for a typical rod pigment. This result is in line with the proposed interpretation that the pre-exponential factor of the theory correlates with the accessibility of the chromophore binding pocket rather than some other features typical of cone pigments.

Although the paper provides only one data point to test the theory, this can be seen as a very important contribution to the field. This is because of the nature of the model relying on multi-vibrational-mode statistical mechanics of visual pigment activation. Direct experimental testing of such molecular properties on a selection of pigments with different spectral properties seems very difficult. Instead, an approach testing the key predictions of the model by careful selections of pigments seems effective. This is exactly what has been done. In addition, the physiological experiments have been carefully carried out with the highest technical standards. Overall, this is a very important contribution to the field. I have only one important technical concern that needs to be addressed by the authors (see below).

Justification of the chromophore exchange rate argument: The authors use microspectrophotometry (MSP) and 9-cis retinal to estimate the exchange rate of the chromophore in E122Q- and WT rods. The authors claim that within the measured times there is no spectral shifts in E122Q-rods that have been exposed to 9-cis retinal. This data is used to conclude that the exchange rate is in line with rod pigments in E122Q-rods. The conclusion is central for the paper. It would be important to validate this argument by giving an estimate of the expected spectral shifts in MSP measurements for a chromophore exchange rate that is in line with that expected for cone-pigments in a similar time interval and relate this predicted shift to the accuracy of the λ_max_ measurements. This is needed to assess to which extent the MSP with its current measurement accuracy for λ_max_ is sensitive to the chromophore exchange rates representing rod and/or cone type pigments on these time scales. This is especially true for E122Q rods where the bleached native rods that have been regenerated with 9-cis retinal have a smaller spectral shift (λ_max_ shift) compared to the WT rods.

*Reviewer #3:*

This paper provides confirmation for a previously published and validated theory of the relationship between the frequency of spontaneous isomerization of a vertebrate photopigment and it wavelength of maximum absorbance- also the peak wavelength of its photoisomerization action spectrum. Although it provides a single data point for this theory, it is important because the theory is more challenging to test for rods (for which spontaneous isomerization is arguably of much greater biological significance than for cones) than for cones because of the limited range of λ_max_ values found in naturally occurring rod pigments. In addition, the frequency is so low for rods that it can only be measured in intact rods, making use of their ability to use extremely efficient amplification mechanisms to allow measurement of single-molecule isomerization events. Making use of an existing mouse model with a knock-in of a rod pigment with a single point mutation that results in a significant blue shift, the authors have quantitatively confirmed that the spontaneous rate is predicted by the theory. Secondarily, they verify that the binding pocket for the chromophore is very similar to that in WT rhodopsin, based on its extremely slow chromophore exchange kinetics.

---

## [Author Response]

*[…] Reviewer #1:*

*Yue et al. here attempt to test an earlier "macroscopic physicochemical theory" (Luo et al., 2011) concerning the basis for thermal noise in visual photoreceptor cells based on an analysis of molecular pigment type (rod versus cone) and wavelength of maximal absorption (λ_max_). The primary experimental system is a knockin/knockout mouse (Rho^E122Q/E122Q^, Gcaps^-/-^) in which the usual wild-type rhodopsin (Rho) is replaced by the mutant Rho E122Q, which has been characterized in some detail earlier using a variety of methods. Much of the paper seems to be a reiteration of the earlier 2011 paper and the authors repeatedly refer to "our" theory or model even though it appears that only 2 of the 8 current authors were associated with the earlier work. Although this paper ascribes to being quantitative in presenting Rho E122Q is a definitive experimental example to validate the earlier theory paper, the claims of the paper are either too vague (for example, using ill-defined terms like "openness" of the retinal binding pocket), or too overreaching (for example, claiming that a single test case proves the earlier theory. In fact, it's not clear that the Rho E122Q mouse is a logical or sufficient model system for a variety of reasons. However, the data set is interesting and valuable and there are no major issues, but the authors should consider rewriting sections of the paper and should address a number of questions/concerns as outlined below:*

We thank this reviewer for commenting that “our data set is interesting and valuable and there are no major issues”. We have now tightened the writing by defining more explicitly what we mean by the “openness” of the retinal-binding pocket. Regarding the choice of the *Rho_E122Q/E122Q_*mouse and only this line for study, the simple reason is that this mouse line happens to be especially interesting by having an exceptionally large λ_max_ shift as well as having hybrid rhodopsin/cone- pigment properties in its mutant rhodopsin.

*1) Abstract. "…spontaneously activated by internal thermal energy" – what does that mean, actually? Isn't is just thermal activation? "Confusion in the field"? What is the confusion? "Quantitatively correlated with the closedness of its chromophore-binding pocket"? How is closedness actually quantitated? It isn't, so any correlation cannot be quantitative. Is it the "closedness" or the λ_max_ value in the end that's the most important factor. The Abstract is extremely confusing in and of itself.*

We like the phrase "spontaneously activated by internal thermal energy" because it introduces the word “spontaneous”, which in the field of visual pigments and phototransduction is understood to refer to the activity of single rhodopsin molecules in the absence of light stimulation, leading to electrical events identical to single-photon responses in both waveform and amplitude. At the same time, we would like to equate the concepts of “spontaneous activity” and “thermal activity”, which may not be obvious to everyone.

"Confusion in the field". The confusion for 30 years is whether the spontaneous activity of a pigment comes from thermal isomerization or from some other irrelevant chemical reaction. This confusion is now more explicitly stated in the revised manuscript. We now define operationally “openness” and “closedness” of the chromophore-binding pocket in a pigment simply by whether the pigment in live cells allows dark-chromophore exchange or not within an experimentally manageable yet appropriately long time frame (3-6 hours). The new data on live zebrafish cones and rods (with the long time required for acquiring these data explaining the delay in submitting the revised manuscript) suggest that, as in the case of salamander red cone pigment previously studied by us (Kefalov et al., Neuron, 2005), there is a clear difference between native rhodopsin and cone pigments with respect to dark chromophore-exchange – namely, either no observable exchange up to 6 hours for rod pigment or >50% exchange already in 2 hr or less for cone pigments. Thus, we have taken the simple approach of referring to these pigments as having “closed” and “open” chromophore-binding pocket, respectively. Should we later find that the openness/closedness of the chromophore-binding pocket ought to be quantified in a finer fashion, we shall refine our theory accordingly. Finally, we would like to emphasize that, in our theory, both λ_max_ and the openness/closedness of the chromophore-binding pocket contribute to the absolute rate of spontaneous activity – with relative importance depending on the absolute λ_max_ value – albeit difficult to elaborate in the Abstract given the required brevity.

*2) Introduction. Of course the ground-state isomerization energy barrier is closely related to the λ_max_. That's trivial, isn't it? Why is that a part of a unifying theory? The trivial definition of "open" pocket refers to accessibility of the Schiff base to hydrolysis, for example by hydroxylamine (with a 1955 Wald reference), but the actual chemistry of the hydrolysis reaction is a very complicated pH dependent mechanism that is only partly dependent on accessibility or hydration state. It is not stated why the E122Q pigment is cone-like, or "non-canonical." Some detail is required to justify its use, other than the mouse model happened to exist. What other mutants might be useful and why weren't they used or considered?*

According to classical physics, a photon with energy whether equal to or exceeding the ground-state isomerization energy barrier should have an equal probability of being absorbed. Thus, the existence of a λ_max_ – namely, the probability of light absorption actually goes down at wavelengths shorter than λ_max_ [though of even higher energy] – does not strictly speaking arise in classical physics, but is a quantum- mechanical concept. In Luo et al., Science, 2011, we have found *experimentally* that a pigment’s λ_max_ is related to its isomerization barrier height, *Ea*, by *Ea* ≈ 0.84*hc*/*λ_max_*, thus allowing calculations in our theory.

Regarding the chromophore-binding pocket, please refer to our answer to Point 1 above. In this revision of the manuscript, we have decided not to use the susceptibility of a pigment to attack by external hydroxylamine as another criterion for the binding pocket’s openness because, from reading the literature, this property appears to depend on experimental conditions and possibly hydroxylamine concentration, thus not a reliable indicator.

The E122Q-rhodopsin is labeled “cone-pigment-like” because it has photochemical kinetics much faster than that of rhodopsin (therefore resembling cone pigments). In this revision, we have dropped the word “canonical” in most instances when describing a rod or cone pigment because we agree that the word is ambiguous. Finally, if pigment mutations *could* lead easily to substantial shifts in λ_max_ or changes in the binding pocket’s openness, and if mouse lines associated with such mutations *were* available, we *would have loved* to measure the spontaneous noise of each and every pigment mutant and check against our theory. However, the truth of the matter is that mutations leading to substantial λ_max_ shifts are rare, based on existing literature. Even more significantly, a large number of rhodopsin mutations lead to rod degeneration, rendering them difficult to study. Thus, it is indeed a good fortune that E122Q-rhodopsin produces a substantial λ_max_ shift and causes no degeneration, and the associated mouse line is available to us for study.

*3) Results, third paragraph. What is the total number of E122Q pigment molecules per cell? Can it be assumed that it is the same as the total Rho in WT? If so, why? The scaling factor related to the number of molecules is very important in the equation used to a spontaneous activation rate constant. Also at the end of the subsection “Measurements of the rates of spontaneous activation”, it is stated that the "molecular rate constant could be obtained by further dividing the measured cellular rate by the number of pigment molecules per rod." Again, how is it assumed that E122Q is the same as Rho?*

E122Q-rhodopsin is expressed at normal level in *Rho_E122Q/E122Q_*mice (Imai et al., J. Biol. Chem., 2006). In the *Gcaps_-/-_*background, we have confirmed the same by Western blotting (Figure 1, please see also response to comment8). Additionally, we have found with single-cell recordings and microspectrophotometry that the pigment contents of *Rho_WT/WT_;Gcaps_-/-_*and *Rho_E122Q/E122Q_;Gcaps_-/-_*rods indeed roughly agree (see revised manuscript).

4) Results, last paragraph. The competition experiment seems problematic. Why was only a single time point of 3 hours in darkness chosen? Does the retinal bind to the 1% BSA predominantly (subsection “Chromophore-preparation”, Methods)?

In Figure 2 single time point of 3 hours in darkness was chosen just for simplicity because, based on previous work from salamander, we think 3 hours should already approach the end point of chromophore exchange if exchange indeed takes place. We have now also added 6 hours in darkness, and saw no difference in result for E122Q- rhodopsin.

Regarding BSA, yes, it is commonly used as a carrier for introducing retinal into rods and cones. Retinal delivery by 1% BSA has been used successfully to restore the sensitivity of bleached photoreceptors within a couple of minutes.

*5) Results, end of last paragraph. It is now stated that E122Q is resistant to hydroxylamine? What? Isn't the λ_max_ of 9-cis Rho 490 nm, not 482 nm? The data actually show Rho at about 498 nm or so, not 500. Why not use the actual data values in the text?*

We have now decided not to include the phenomenon of susceptibility of a pigment to attack by external hydroxylamine as another criterion for the binding pocket’s openness because, from reading the literature, this property appears to depend on experimental conditions and possibly hydroxylamine concentration, thus not a reliable indicator.

From the literature, the λ_max_ of 9-*cis*-rhodopsin is in the range of 481-489 nm. The λ_max_ for WT rhodopsin (with 11-cis retinal) is about 500 nm, but can also vary a bit in different hands and also in the literature. Our measured λ_max_’s of both 11-*cis*- and 9- *cis*-rhodopsin fall within these ranges. The key point of the experiment here is that there is either a shift or no shift in λ_max_ after dark incubation with 9-*cis*. Thus, the exact λ_max_ value is not critical. We have, however, followed the reviewer’s suggestion of using our own measured λ_max_ (481 nm for E122Q-rhodopsin) instead of a previously reported value (480 nm from Imai et al., J. Biol. Chem., 2006) for theoretical noise prediction.

*6) Discussion. The main criticism is that the authors need to justify how one example case can provide a justification for the earlier theory, and why and how the E122Q Rho is a legitimate test system. Does it really have hybrid rod-cone properties when expressed in a mouse rod cell?*

Please see the final paragraph in our response to point 2 above.

We would like to emphasize that E122Q-rhodopsin represents an important experimental model because: i) its λ_max_ is considerably blue-shifted, thus providing a valuable data point for rod pigments, which normally have a very limited range of λ_max_, and ii) it has fast photochemical kinetics characteristic of cone pigments (Imai et al., J. Biol. Chem., 2006) but, on the other hand a rod-pigment-like chromophore-binding pocket, thus allowing the teasing apart of the contributions of pocket openness and of other cone-pigment properties to the higher noise of cone pigments. We have now modified the text to make these points clearer.

*7) Subsection “Pigment noise prediction”. m is taken to be the same for Rho and cone pigments? That's really strange and some additional explanation seems warranted.*

We used the same *m* value for both rhodopsin and cone pigments, as we did previously (Luo et al., Science, 2011) because they have the same chromophore. Given that the isomerization reaction occurs in the chromophore, it is not unreasonable to think that the energy for thermal isomerization does come mostly, if not all, from the chromophore itself.

*8) Figure 1. Again, referring to the issue of E122Q quantitation, the Rho band looks much heavier than the E122Q knockin band.*

In Figure 1, the left two lanes are duplicates of *Rho_WT/WT_;Gcaps_-/-_*retinas and the right two lanes are duplicates of *Rho_E122Q/E122Q_;Gcaps_-/-_*retinas. On the whole, there is no significant difference between the two genotypes. Moreover, single-cell recordings and microspectrophometry suggest that the rhodopsin content is comparable between the two genotypes (see text of revised manuscript).

Reviewer #2:

*[…] Justification of the chromophore exchange rate argument: The authors use microspectrophotometry (MSP) and 9-cis retinal to estimate the exchange rate of the chromophore in E122Q- and WT rods. The authors claim that within the measured times there is no spectral shifts in E122Q-rods that have been exposed to 9-cis retinal. This data is used to conclude that the exchange rate is in line with rod pigments in E122Q-rods. The conclusion is central for the paper. It would be important to validate this argument by giving an estimate of the expected spectral shifts in MSP measurements for a chromophore exchange rate that is in line with that expected for cone-pigments in a similar time interval and relate this predicted shift to the accuracy of the λ_max_ measurements. This is needed to assess to which extent the MSP with its current measurement accuracy for λ_max_ is sensitive to the chromophore exchange rates representing rod and/or cone type pigments on these time scales. This is especially true for E122Q rods where the bleached native rods that have been regenerated with 9-cis retinal have a smaller spectral shift (λ_max_ shift) compared to the WT rods.*

Prompted in part by the comments of this reviewer, we have taken the time to study another animal species, zebrafish, which has multiple cone types with distinct pigments. The zebrafish data are now included in the revised manuscript. The bottom line is that, to the extent resolvable, the property of dark chromophore-exchange does appear to be common across cone pigments but negligible in rhodopsin. Regarding the chromophore-binding pocket, we now simply define it operationally as being “open” if the pigment shows the property of dark chromophore exchange with an exogenous chromophore without isomerization. Rhodopsin shows no sign of any exchange up to 6 hours, whereas cone pigments show substantial exchange in the order of an hour, whether in salamander (Kefalov et al., Neuron, 2005) or zebrafish (in this paper). We would like to emphasize here (and have stated as such in the paper) that our work, which uses one value for the pre-exponential factor *A* for rod pigments and another for cone pigments (i.e., a binary situation), is not necessarily intended to be the final word. If it turns out in the future that the *A* value has more than two possible values, we may have to fine-tune the theory.

*Reviewer #3:*

This paper provides confirmation for a previously published and validated theory of the relationship between the frequency of spontaneous isomerization of a vertebrate photopigment and it wavelength of maximum absorbance- also the peak wavelength of its photoisomerization action spectrum. Although it provides a single data point for this theory, it is important because the theory is more challenging to test for rods (for which spontaneous isomerization is arguably of much greater biological significance than for cones) than for cones because of the limited range of λ_max_ values found in naturally occurring rod pigments. In addition, the frequency is so low for rods that it can only be measured in intact rods, making use of their ability to use extremely efficient amplification mechanisms to allow measurement of single-molecule isomerization events. Making use of an existing mouse model with a knock-in of a rod pigment with a single point mutation that results in a significant blue shift, the authors have quantitatively confirmed that the spontaneous rate is predicted by the theory. Secondarily, they verify that the binding pocket for the chromophore is very similar to that in WT rhodopsin, based on its extremely slow chromophore exchange kinetics.

We thank this reviewer for the general compliments about our work.